# Modeling Behavior of *Salmonella* spp. and *Listeria monocytogenes* in Raw and Processed Vegetables

**DOI:** 10.3390/foods13182972

**Published:** 2024-09-19

**Authors:** Su Bin Son, Ha Kyoung Lee, So Jeong Kim, Ki Sun Yoon

**Affiliations:** Department of Food and Nutrition, College of Human Ecology, Kyung Hee University, 26 Kyungheedae-ro, Dongdaemun-gu, Seoul 02447, Republic of Korea; sbson@khu.ac.kr (S.B.S.); lik0922@naver.com (H.K.L.); thwjd8836@naver.com (S.J.K.)

**Keywords:** predictive modeling, microbial growth, *Salmonella* spp., *Listeria monocytogenes*, raw and processed vegetables

## Abstract

Given the persistent occurrence of foodborne illnesses linked to both raw and processed vegetables, understanding microbial behavior in these foods under distribution conditions is crucial. This study aimed to develop predictive growth models for *Salmonella* spp. and *Listeria monocytogenes* in raw (mung bean sprouts, onion, and cabbage) and processed vegetables (shredded cabbage salad, cabbage and onion juices) at various temperatures, ranging from 4 to 36 °C. Growth models were constructed and validated using isolated strains of *Salmonella* spp. (*S*. Bareilly, *S.* Enteritidis, *S*. Typhimurium) and *L. monocytogenes* (serotypes 1/2a and 1/2b) from diverse food sources. The minimum growth temperatures for *Salmonella* varied among different vegetable matrices: 8 °C for mung bean sprouts, 9 °C for both onion and cabbage, and 10 °C for ready-to-eat (RTE) shredded cabbage salad. Both pathogens grew in cabbage juice at temperatures above 17 °C, while neither demonstrated growth in onion juice, even at 36 °C. Notably, *Salmonella* spp. exhibited faster growth than *L. monocytogenes* in all tested samples. At 8 °C, the lag time (LT) and specific growth rate (SGR) for *Salmonella* spp. in mung bean sprouts were approximately tenfold longer and threefold slower, respectively, compared to those at 10 °C. A decrease in refrigerator storage temperature by 1 or 2 degrees significantly prevented the growth of *Salmonella* in raw vegetables. These findings offer valuable insights into assessing the risk of foodborne illness associated with the consumption of raw and processed vegetables and inform management strategies in mitigating these risks.

## 1. Introduction

In response to growing concerns about animal rights and environmental sustainability, consumer interest in “green dining tables” is increasing [1]. This trend is particularly evident in the heightened demand for fresh convenience foods, as demonstrated by the popularity of ready-to-eat (RTE) salads [2]. According to the Korea Rural Economic Institute’s report about “Changes in the market for fresh convenience fruits and vegetables and response tasks” [3], vegetables accounted for nearly 90% of the total 70,730 tons of fresh convenience fruits and vegetables produced in 2018. Notably, fresh-cut onions (6654 tons) and fresh-cut cabbage (4420 tons) were the primary products. A recent survey by Son et al. [2] in Korea found that onions are the most favored raw vegetable, with 36.9% of respondents preferring them. Cabbage, on the other hand, has the highest purchase rate among fresh-cut vegetables at 53.3% and is primarily consumed as salads. The survey, which involved 1,201 respondents, also revealed that 34.5% consume vegetables in their raw form, followed by consumption in salad form (23.8%), as cooked vegetables (22.1%), raw with seasoning (13.1%), and finally, in grated or juiced form (6.3%). Moreover, the domestic sales of fresh convenience foods, specifically in the sprouts category, surged from 9780 USD in 2020 to 105,186 USD in 2021 [4], reflecting a notable increase in consumption. In Korea, a survey of 1201 respondents revealed that 30.4% consumed raw sprouts [2].

As the domestic and global consumption of fresh vegetables rises, so does the incidence of food-borne illness outbreaks linked to these products. Between 2018 and 2023 in the United States, *Salmonella*, *Escherichia coli* O157:H7, *Listeria monocytogenes*, and *Cyclospora* were responsible for 3156 cases and five fatalities. Contaminated onions by *Salmonella Oranienburg* and packaged salads contaminated with *L. monocytogenes* resulted in 1040 cases and three deaths, respectively [5]. Similarly, Europe observed that 34.1% of foodborne illnesses related to vegetable consumption resulted in 5385 cases and 53 fatalities over the past two decades [6]. As of 2023, *Salmonella* was identified as the cause in 45 (12.4%) out of 362 food poisoning outbreaks in Korea, accounting for 2419 (28.5%) of the 8485 reported cases [7]. Jeon et al. [8] found that *L. monocytogenes* tested positive (25–50%) in various ingredients such as beef, mung bean sprouts, lettuce, cabbage, and pumpkin, which are commonly included in Shabu-Shabu meal kits sold both online and offline in Korea.

Foodborne illness linked to vegetable sprouts has been reported in many countries, with recalls often due to *Salmonella* spp. and *L. monocytogenes* [9,10]. In 2018, 64 people in Korea were infected by seasoned mung bean sprouts and crab sticks contaminated with *L. monocytogenes* [11]. Given the continuous reports of food-borne illnesses associated with fresh vegetables, it is crucial to predict microbial behavior in commonly consumed raw and processed vegetables during distribution conditions for risk management.

Recent research on the development of predictive models for foodborne pathogens has focused on various produce items, targeting specific pathogens. Studies have examined leafy greens [12], germinating alfalfa seeds [13], fresh-cut iceberg lettuce [14,15], whole cucumbers [16], and sliced cucumbers [17], specifically for *Salmonella* spp. Similarly, predictive models for *L. monocytogenes* have been developed for leafy greens [12], fresh-cut fruits and vegetables [18], and fresh-cut cucumbers [19].

Despite global efforts in predictive modeling, there has been limited research comparing the likelihood of food poisoning caused by *Salmonella* and *Listeria* across various types of raw and processed vegetables in Korea. This lack of comprehensive research is particularly concerning given the recent rise in the consumption of RTE fresh vegetables and salads in the domestic market. Addressing these gaps is crucial for developing more effective food safety strategies and ensuring public health.

In a previous study [20], a semi-quantitative risk assessment tool called Risk Ranger was employed to assess the potential risks of *Salmonella* and *L. monocytogenes* in 13 different vegetables: onion, green leafy vegetables, Napa cabbage, cabbage, tomato, cucumber, iceberg lettuce, spinach, paprika, mung bean sprouts, sprout vegetables, celery, and melon. Following this, we reviewed a survey study [2] that analyzed vegetable purchase and consumption habits, along with perceptions of foodborne illness risks in Korea. Then, onion, cabbage, mung bean sprouts, RTE cabbage salad, and cabbage and onion juices were selected for the current study based on increased consumption rates, popularity, health benefits, and gaps in existing research.

The objective of this study was to develop growth models for *Salmonella* spp. and *L. monocytogenes* that reflect the specific distribution conditions of these vegetables. Additionally, the performance of these models was validated using different strains (isolated from various foods) and at temperatures not initially included in the model development (20 and 30 °C).

## 2. Materials and Methods

### 2.1. Analysis of Microbiological Quality, pH, and Water Activity (aw) of Vegetables

The hygiene indicator bacteria, pH, and aw of mung bean sprouts, onion, cabbage, RTE shredded cabbage salad, and cabbage and onion juice were analyzed. Each 25 g sample was aseptically measured and pummeled using a stomacher (BagMixer, Interscience, St Nom la Bretêche, France) for 2 min with 225 mL of sterile 0.1% peptone water (Becton, Dickinson and Company, Sparks, NV, USA) in a filter bag (F-78860, Interscience). One mL aliquot of the diluted solution was inoculated onto 3M Petrifilm^TM^ aerobic count (AC) and *E. coli*/coliform (EC) plates (3M Corporation, St. Paul, MN, USA), which were incubated at 36 ± 1 °C for 48 and 24 h, respectively. Colonies were counted using a 3M Petrifilm^TM^ Plate Reader Advanced (3M PPRA; 3M corporation, St. Paul, MN, USA).

Each 10 g of raw and processed vegetables was homogenized with 90 mL of distilled water using a stomacher for 2 min, and the pH values were measured in triplicate with a pH meter (Orion^TM^ Star A211, Thermo Fisher Scientific Co., Waltham, MA, USA). The aw of samples (15 g) was also measured in triplicate using a water activity analyzer (Rotronic HP23-AW-A, Rotronic AG, Grindelstrasse, Bassersdorf, Switzerland).

### 2.2. Bacterial Strain Preparation for Predictive Model Development

Various standard and isolated strains from foods were employed for model development and validation. Initially, *S*. Typhimurium (ATCC 13311), *S.* Enteritidis (KCCM 12021), and *L. monocytogenes* strains isolated from enoki mushrooms (serotype 1/2b) and smoked salmon (serotype 1/2a) were used. Later in the study, S. Bareilly (isolated from carrots and soybean sprouts) and *S.* Enteritidis (isolated from iceberg lettuce) as well as isolated *L. monocytogenes* strains from lettuce (serotype 1/2b) and vegetables (serotype 1/2a), obtained from the Ministry of Food and Drug Safety (Cheongju-si, Chungcheongbuk-do, Republic of Korea), were used.

Each strain was maintained at −80 °C in brain heart infusion broth (BHI; Oxoid, Basingstoke, Hampshire, UK) and tryptone soya broth (TSB; Oxoid) with 0.6% yeast extract containing 20% glycerol (Ducksan Pure Chemicals Co., Ltd., Ansan, Gyeonggi-do, Republic of Korea), respectively. Thawed cultures of *Salmonella* spp. and *L. monocytogenes* (10 µL each) were inoculated into 10 mL BHI broth and TSB containing 0.6% yeast extract, respectively. These cultures were incubated at 36 ± 1 °C for 24 h in a rotary shaker (VS-8480SP, Vision Scientific, Daejeon, Republic of Korea) at 140 rpm. After incubation, the cultures were transferred to 50 mL conical tubes and centrifuged (VS-550, Vision Scientific) at 4000 rpm for 10 min. The supernatant was removed, and the cell pellets were resuspended and washed twice with 10 mL of 0.1% peptone water. The resuspended pellets of each strain were then aseptically combined into a 50 mL conical tube to prepare the cocktail inoculum. This inoculum was further diluted serially with sterile 0.1% peptone water for use in the experiments.

### 2.3. Sample Preparation and Inoculation Strains for Model Development

Mung bean sprouts, onions, and cabbage were purchased from a large supermarket (Seoul, Republic of Korea). Additionally, RTE shredded cabbage salad (Just Green Agricultural Corp., Anseong, Gyeonggi-do, Republic of Korea), organic cabbage juice (Hwangji Inc., Nonsan, Chungcheongnam-do, Republic of Korea), and organic onion juice (Jangsu Story Agricultural Corp., Gimpo, Gyeonggi-do, Republic of Korea) were purchased online through Naver Shopping and used within 24 h. These processed products were selected based on criteria such as popularity, sales rates, sample supply stability, and HACCP certification. The RTE shredded cabbage salad underwent multiple processing steps, including peeling, shredding, washing, disinfection with peracetic acid, and packaging in modified atmosphere packaging (Hypervac, Suwon, Gyeonggi-do, Republic of Korea). As a result, the preparation process for this product differs from that of raw cabbage. Both the cabbage juice and onion juice are freshly squeezed products, free of preservatives and colorants, and are sterilized and packaged individually in polyethylene and polypropylene pouches, respectively.

The inoculation experiment was conducted in a sterile environment using a clean bench. All stainless-steel instruments, such as knives, forceps, trays, and scissors, were sterilized in advance using an autoclave. Onion and RTE shredded cabbage samples were used without pretreatment. For the onion samples, the outer peel and core were aseptically removed, and the onions were sliced to a thickness of 0.5 cm. Ten-gram portions were placed in petri dishes and inoculated with 100 µL of a mixed suspension of *Salmonella* spp. and *L. monocytogenes* on the sample surfaces.

For cabbage and mung bean sprouts, a pre-treatment process was conducted to remove the background microorganisms before inoculation of *Salmonella* spp. The outer leaves and core of the cabbage were aseptically removed and then sliced into 0.5 cm thicknesses. Subsequently, the sliced cabbage and mung bean sprouts were immersed in 3.6% hydrogen peroxide (H_2_O_2_; Meditop, Yongin, Gyeonggi-do, Republic of Korea) for 5 min [21], followed by a 2 min wash in sterile distilled water. Afterwards, they were further irradiated for 30 min under a UVC low-pressure mercury lamp (Sankyo G40T10, Sankyo Denki Co., Hiratsuka, Kanagawa, Japan). In the case of mung bean sprouts, despite the preprocessing mentioned above, the presence of residual microbes hindered the development of a predictive growth model for *Salmonella*. Therefore, an additional preprocessing step was conducted; the pre-treated mung bean sprouts were additionally blanched at 100 °C for 13 s using a digital hot plate stirrer (DH.WMH 01500, DAIHAN Scientific Co., Ltd., Wonju, Gangwon-do, Republic of Korea) before inoculation with *Salmonella* spp. The temperature of the sterile distilled water was measured by a thermometer (Hee-sung, Bucheon, Gyeonggi-do, Republic of Korea), which was placed in the middle of the water. After pretreatment, the general bacterial count, coliform group, pH, and water activity were measured to verify the effectiveness of the pretreatment process. The samples were used after confirming that the pretreatment process effectively reduced the number of general bacteria and coliforms, and that the essential characteristics of the sample were maintained as there was no significant change in pH and water activity.

After eliminating residual microorganisms, a 7 g portion of pretreated mung bean sprouts was placed in a petri dish and contaminated by spot-inoculating it with 100 µL of each cocktail suspension. Subsequently, 3 g of pretreated sprouts was placed over the contaminated mung bean sprouts to facilitate cross-contamination. The pretreated cabbage samples were portioned into 10 g quantities and placed in petri dishes. The surfaces of the samples were inoculated with 100 µL of a mixed suspension of both *Salmonella* spp. and *L. monocytogenes*.

All inoculated samples were air-dried for 30 min in the clean bench to allow complete absorption of the bacterial suspension. Once absorption was confirmed, the petri dishes were covered with lids and sealed in polyethylene film (Seven L Pack Co., Gwangju, Gyeonggi-do, Republic of Korea) under aerobic conditions. This procedure ensured an initial bacterial load of 1.5 to 2.0 log CFU/g on the sample surfaces. Then, the inoculated samples were stored in the chamber (VS-1203PFC, Vision Scientific) at various temperatures, ranging from 7 to 36 °C for *Salmonella* spp. and from 4, to 36 °C for *L. monocytogenes*. For cabbage juice and onion juice, 30 mL of each sample was placed into a 50 mL conical tube (SPL Life Sciences Co., Pocheon, Gyeonggi-do, Republic of Korea). Each tube was inoculated with 300 µL for each cocktail suspension, vortexed, and stored at temperatures of 17, 25, and 36 °C.

During storage, at regular intervals and depending on the temperature, 10 g samples of mung bean sprouts, onions, cabbage, and cabbage salad were homogenized in 90 mL of sterilized 0.1% peptone water. The initial sample, taken at 0 h, was analyzed from the moment the samples were packaged. Similarly, 1 mL each of cabbage juice and onion juice was used as the test sample. These homogenized samples were then serially diluted in 0.1% peptone water and inoculated onto Xylose Lysine Desoxycholate (XLD, Oxoid) agar for *Salmonella* spp., and onto Polymyxin Acriflavin LiCI Ceftazidime Esculin Mannitol (PALCAM, Oxoid) agar for *L. monocytogenes*. The inoculated agars were incubated at 36 ± 1 °C for 24 h and 48 h, respectively. Colony counts for each pathogen were performed using a colony counter (Scan 1200, Interscience).

### 2.4. Model Development and Validation of Model Performance

The primary models for *Salmonella* spp. and *L. monocytogenes* in raw and processed vegetables were developed as a function of time at each temperature using the Modified Gompertz model (V7.03, GraphPad software, San Diego, CA, USA) (Equation (1)). The growth kinetic parameters, which include the lag time (LT), the maximum specific growth rate (SGR), and the maximum population density (MPD) were calculated [22].
Modified Gompertz model: Y = N_0_ + C × exp[−exp{(2.718 × SGR/C) × (LT − X) + 1}](1)

N_0_: log initial number of bacteria (log CFU/g)C: difference between initial and final bacteria numbers (log CFU)LT: lag time (h)SGR: specific growth rate (log CFU/h)X: time (h)Y: log number of bacteria (log CFU/g)

For the development of the second predictive model, which evaluates the effect of storage temperature on the parameters (LT, SGR, MPD) of the primary models, the Davey model [23] (Equation (2)), the square root model [24] (Equation (3)), the second-order polynomial model [25] (Equation (4)), and the third-order polynomial model [26] (Equation (5)) were applied. The secondary models were evaluated by comparing their goodness of fit (R^2^) with the parameters of the primary model. Those secondary models demonstrating superior accuracy and predictive performance, as indicated by the highest goodness of fit values, were identified and selected as the optimal models for the study.
Davey model: Y = a + (b/T) + (c/T^2^)(2)
Y: lag time (h)a, b, c: constantsT: temperature (°C)
Square root model: Y = {b(T − T_min_)}^2^(3)
Y: specific growth rate (log CFU/h)b: constantT: temperature (°C)T_min_: theoretical minimum temperature (°C)
Second-order polynomial model: Y = b_0_ + (b_1_ × T) + (b_2_ × T^2^)(4)
Y: maximum population density (log CFU/g)b_0_, b_1_, b_2_: constantsT: temperature (°C)
Third-order polynomial model: Y = b_0_ + (b_1_ × T) + (b_2_ × T^2^) + (b_3_ × T^3^)(5)
Y: maximum population density (log CFU/g)b_0_, b_1_, b_2_, b_3_: constantsT: temperature (°C)

To validate the performance of predictive models for *Salmonella* spp. and *L. monocytogenes* in raw and processed vegetable products, temperatures not previously used in model development (20 °C and 30 °C) were tested. These tests involved either the same strains used during model development (interpolation) or different strains that were not part of the original model development process (extrapolation). For validating the *Salmonella* spp. model, strains isolated from carrots (*S*. Bareilly), soybean sprouts (*S*. Bareilly), and iceberg lettuce (*S.* Enteritidis) were used. Additionally, during the study period, S. Typhimurium was isolated from cabbages obtained from the market and was used to validate the model for RTE shredded cabbage salad. In the case of the *L. monocytogenes* model, strains from lettuce (1/2b), vegetables (1/2a), enoki mushrooms (1/2b), and smoked salmon (1/2a) were also used. Different strains were used alternately during model development and validation. The accuracy of the predictive model for *Salmonella* spp. and *L. monocytogenes* was also cross-verified by different personnel.

The performance of these models was then evaluated using the adjusted coefficient of determination(R^2^), root mean square error (RMSE) (Equation (10)), bias factor (B*_f_*) (Equations (6) and (8)) to indicate relative deviation, and accuracy factor (A*_f_*) (Equations (7) and (9)) to measure prediction accuracy. These metrics were employed to evaluate the appropriateness of the predictive models [27,28,29]. B*_f_* gives a fail-safe model (fail-safe, B*_f_* < 1; fail-dangerous, B*_f_* > 1), which evaluates the relative standard deviation between the predicted and observed, and the values at 0.7~1.15 represent the safe model. A*_f_* represents the accuracy of the model and when the value is closer to one, it is the most accurate model. RMSE estimates the suitability and is a proper model when the value is closer to zero.
B*_f_* for LT = 10^∑log(predicted/observed)/n^(6)
A*_f_* for LT = 10^∑|log(predicted/observed)|/n^(7)
B*_f_* for SGR and MPD = 10^∑log(observed/predicted)/n^(8)
A*_f_* for SGR and MPD = 10^∑|log(observed/predicted)|/n^(9)
(10)$$\mathrm{RMSE}=\sqrt{\frac{1}{n}\times \sum {\left(\mathrm{observed}\;\mathrm{value}-\;\mathrm{predicted}\;\mathrm{value}\right)}^{2}}$$
n: total number of experimental values obtained from independent variables or predictive values obtained from the developed growth model.

### 2.5. Statistical Analysis

All experiments were conducted in duplicate, with two replicate samples used per experiment, resulting in a total of four replicates (*n* = 4) for each sample across the two experiments. Growth model fitting was performed using GraphPad Prism Software, and growth kinetics parameters (LT, SGR, and MPD) were obtained. The mean and standard deviation of LT, SGR, and MPD were calculated based on the four replicates to ensure data consistency and reliability for subsequent statistical analysis. Significant differences in pathogen growth kinetics across different vegetables and temperatures were analyzed using one-way analysis of variance (ANOVA) followed by Duncan’s multiple range tests, conducted with SAS software version 9.4 (SAS Institute, Inc., Cary, NC, USA). Data are presented as mean ± standard deviation, and statistical significance was set at *p* < 0.05.

## 3. Results and Discussion

### 3.1. Analysis of Microbiological Quality, pH, and Water Activity (aw)

The hygiene indicator bacteria, pH, and aw of raw and processed vegetables were analyzed to characterize their properties before developing predictive models (Table 1). The analysis showed that mung bean sprouts had the highest level of total aerobic bacteria (TAB) at 7.23 log CFU/g, followed by cabbage at 4.67 log CFU/g, onions at 2.73 log CFU/g, and RTE shredded cabbage salad at 2.02 log CFU/g. Total coliform counts (TCC) were absent in all samples except mung bean sprouts, which had a significant count of 6.03 log CFU/g. The populations of TAB and TCC in various types of sprouts, such as broccoli, clover, soybeans, and alfalfa, ranged from 6.8 to 9.7 log CFU/g and 5.0 to 8.8 log CFU/g, respectively [30]. Additionally, studies by Waje et al. [31] and Park et al. [32] consistently found that TAB counts in mung bean sprouts exceeded 7 log CFU/g. These high levels of TAB explain the rapid spoilage of mung bean sprouts, even under refrigeration. Notably, *E. coli* was not detected in any of the raw or processed vegetables tested in this work. In a previous work [20], samples were systematically collected and analyzed on a seasonal basis over a period of two years (2022~2023), considering the production rate from different areas. Out of a total of 2520 samples, neither pathogen was detected, except in cabbage. S. Typhimurium was found in six out of 360 cabbage samples (1.67%) obtained from both online and offline markets.

The average pH and aw values for these vegetables were 6.01 and 0.956, respectively. These conditions are generally ideal for the growth of most microorganisms, highlighting the need for strict hygiene practices in handling, including washing, sanitizing, and storing raw vegetables, specially RTE vegetables without temperature abuse.

### 3.2. Primary Growth Models of Salmonella spp. and L. monocytogenes

The growth kinetics of *Salmonella* spp. and *L. monocytogenes* in raw and processed vegetables were evaluated across a range of storage temperatures, chosen to cover the minimum growth temperature range for each microorganism. For *Salmonella* spp., temperatures of 7, 8, 9, 10, 15, 17, 25, and 36 °C were investigated (Table 2), whereas for *L. monocytogenes*, the temperatures studied were 4, 10, 17, 25, and 36 °C (Table 3).

#### 3.2.1. *Salmonella* spp. Growth in Raw and Processed Vegetables

The growth of *Salmonella* spp. was not observed in mung bean sprouts at temperatures below 8 °C, in raw onion and cabbage below 9 °C, in RTE shredded cabbage salad below 10 °C, and in cabbage juice below 17 °C (Table 2). These findings demonstrate that maintaining specific storage temperatures effectively inhibits the growth of *Salmonella* spp. in both raw and processed vegetables. Notably, at 8 °C, the LT for *Salmonella* spp. in mung bean sprouts was approximately 10 times longer, and the SGR was three times slower compared to those at 10 °C. This finding highlights the significant impact that even a slight reduction in refrigerator storage temperature can have on enhancing the microbiological safety of mung bean sprouts.

The LT for *Salmonella* spp. in raw onions and cabbage at 9 °C was 169.5 h and 292.4 h, respectively. This indicates that *Salmonella* spp. requires a longer adaptation period in cabbage compared to onions, suggesting a more resistant inhibitory environment in cabbage. This resistance in cabbage can be attributed to factors such as its lower water activity (aw) and higher TAB counts compared to onions (Table 1). Furthermore, in cabbage, an increase in storage temperature from 9 °C to 10 °C resulted in a more than fourfold decrease in the LT of *Salmonella* spp. (Table 2). However, this temperature change did not significantly affect the SGR, indicating that while the bacteria adapted faster at slightly higher temperatures, their growth rate remained relatively stable. The findings of this study demonstrate that even a small temperature difference of approximately 1 °C can significantly extend the LT of *Salmonella* spp. in cabbage. This suggests that even minor adjustments in storage temperatures can significantly influence the initial adaptation phase of *Salmonella* spp., potentially affecting the safety of stored cabbage by either delaying or accelerating bacterial growth. Overall, the SGR of *Salmonella* spp. was found to be higher in onions compared to cabbage, indicating that *Salmonella* poses a greater risk in onions than in cabbage.

While *Salmonella* spp. exhibited growth in cabbage stored at 9 °C, it did not grow in RTE shredded cabbage salad stored at the same temperature. Additionally, the LT of *Salmonella* spp. in RTE shredded cabbage salad was approximately 1.8 times longer than in raw cabbage at 10 °C. The MPD of *Salmonella* spp. in RTE shredded cabbage salad was 4.76 log CFU/g at 17 °C, lower than the 6.04 log CFU/g observed in raw cabbage. This extended LT and reduced MPD in RTE shredded cabbage salad, compared to raw cabbage, can be attributed to the additional washing and disinfection processes that RTE salad undergoes before distribution.

Since *Salmonella* spp. did not exhibit growth in cabbage juice at temperatures below 17 °C, primary models were developed specifically at 17, 25, and 36 °C to understand better its growth dynamics at higher temperature. At a common storage temperature of 17 °C, the longest LT (42.37 h) was observed in cabbage juice, followed by RTE shredded cabbage (17.11 h), and cabbage (14.95 h). This indicates that cabbage juice provides a more challenging environment for *Salmonella* spp. to initially adapt and begin growing. Furthermore, the SGR for these products at 17 °C showed a clear order: cabbage had the highest SGR at 0.120 log CFU/h, followed by cabbage juice at 0.096 log CFU/h, with RTE shredded cabbage salad exhibiting the slowest rate at 0.058 log CFU/h. This suggests that the processing involved in RTE shredded cabbage salad, likely the washing and disinfection technique using peracetic acid, significantly inhibits the growth of *Salmonella* spp., contributing to its lower growth rate compared to that of fresh cabbage and cabbage juice.

At 25 °C, the MPD of *Salmonella* spp. was also observed to be the lowest in cabbage juice at 4.93 log CFU/mL compared to RTE shredded cabbage salad (5.59 log CFU/g) and cabbage (6.69 log CFU/g), indicating that the growth of *Salmonella* spp. was most prohibited in cabbage juice. This inhibition can be partially attributed to the lower pH level of cabbage juice, which was measured at 5.2, compared to the pH levels of cabbage (6.71) and RTE shredded cabbage salad (6.8) as shown in Table 1. Furthermore, it was found that when the storage temperature increases to 25 °C, the pre-disinfection effectiveness of RTE salad decreases.

Among the tested samples in this work, except onion juice, *Salmonella* spp. demonstrated the most vigorous growth in mung bean sprouts, followed by onions, cabbage, RTE shredded cabbage salad, and cabbage juice in descending order of growth rate. This sequence highlights how different processing and storage conditions impact the proliferation of pathogens in various raw and processed vegetables.

#### 3.2.2. Listeria Growth in Raw and Processed Vegetables

At 4 °C, the most extended LTs for *L. monocytogenes* were recorded in cabbage (371.6 h), onion (363 h), RTE shredded cabbage salad (306.8 h), and mung bean sprouts (83.9 h), as shown in Table 3. As the storage temperature was raised from 4 °C to 10 °C, the LT for *L. monocytogenes* was significantly reduced, approximately 14 to 24-fold in onion, cabbage, and RTE shredded cabbage salad, and 6-fold in mung bean sprouts. At 4 °C, the highest SGRs of *L. monocytogenes* were observed in mung bean sprouts (0.018 log CFU/g), followed by RTE shredded cabbage salad (0.012 log CFU/h), onion (0.011 log CFU/h), and cabbage (0.010 log CFU/h). In the present study, we observed growth of *L. monocytogenes* in mung bean sprouts, onion, and cabbage, as well as in RTE shredded cabbage salad stored at 4 °C, starting approximately on the 15th day of storage. Conversely, no growth of *L. monocytogenes* was detected in fresh-cut salad over a 10-day period at the same temperature [18]. Variations in strains, ingredient compositions, and pretreatments were noted between the two studies. In the research conducted by Alegbeleye et al. [33], only one out of 14 vegetable salads exhibited growth of *L. monocytogenes* at 4 °C. Other RTE shredded cabbage salads showed different bacterial behaviors, influenced by storage temperature and duration, as well as the types of vegetables included in the salad. Notably, even when identical bacterial strains were used, varying proliferation patterns were observed depending on the food matrix. Recently, Feng et al. [19] reported that *L. monocytogenes* demonstrated growth in fresh-cut cucumbers at 5 °C, with a maximum growth rate of 0.0065 log CFU/g/h. The lag time observed was 82 h, and the population reached a maximum of 5.431 log CFU/g.

*L. monocytogenes* did not exhibit growth in commercially processed cabbage juice stored below 17 °C, a pattern also observed for *Salmonella* as shown in Table 2. The antimicrobial properties of methyl methanethiosulfinate (MMTSO), which is likely produced during the juicing and filtering process through enzymatic activity in disrupted cabbage tissue, contributed to this inhibition of *Staphylococcus aureus* [34]. MMTSO can spontaneously generate another potent antimicrobial compound, MMTSO_2_ [35], and has a minimum inhibitory concentration of 50 ppm against *L. monocytogenes* B70 (ATCC 19115) [36]. This compound interacts with the sulfhydryl (-SH) groups on the cell membrane, disrupting physiological functions of cells, and thereby inhibiting bacterial growth [36]. No growth of *L. monocytogenes* was observed in cabbage juice stored at 10 °C. The juice was prepared directly from raw cabbage, which was juiced and filtered [37].

An analysis of the growth behavior of *Salmonella* spp. and *L. monocytogenes* across various raw and processed vegetables revealed a consistent pattern. Notably, *Salmonella* spp. exhibited faster growth than *L. monocytogenes* in all samples tested, with higher MPDs ranging from 3.43 to 9.30 log CFU/g, compared to 2.87 to 8.45 log CFU/g for *L. monocytogenes* at all temperatures. These findings align with previous research; for instance, Ndraha et al. [15] reported that the SGR of *S. enterica* in lettuce stored at 25 °C was higher than that of *L. monocytogenes*. Similarly, *S. enterica* grew faster than *L. monocytogenes* in minimally processed packaged lettuce across temperatures ranging from 5 to 30 °C [38].

In the case of onion juice, neither of these pathogenic bacteria demonstrated growth in onion juice, even at the elevated temperature of 36 °C (Table 2 and Table 3). This resistance to bacterial proliferation is attributed to the low pH of onion juice and the presence of antibacterial substances like quercetin, which is extracted from onions and inhibits the growth of most Gram-positive and Gram-negative bacteria [39]. Among 13 different vegetables surveyed in Korea, including onion, green leafy vegetables, Napa cabbage, cabbage, tomato, cucumber, iceberg lettuce, spinach, paprika, mung bean sprouts, sprout vegetables, celery, and melon, onions have the highest consumption rate. Notably, 36.9% of onion consumption occurs through non-heated methods, such as raw intake. Onions are highly nutritious, rich in beneficial compounds that contribute to health and physiological well-being [40]. Thus, consumer preference for onion juice is on the rise.

### 3.3. Secondary Model and Validation of Model Performance

The secondary models for LT and SGR of *Salmonella* spp. (Figure 1A,B) and *L. monocytogenes* (Figure 2A,B) were constructed using the Davey model and square-roots model, respectively. For the MPD, *Salmonella* spp. (Figure 1C) was modeled using a second-order polynomial, while a third-order polynomial was employed for *L. monocytogenes* (Figure 2C). These models were specifically designed to describe the dynamics of each pathogen in raw and processed vegetables. They account for temperature variations to assess changes in contamination levels, thereby predicting the associated risk for each food-pathogen pair during various distribution conditions (Table 4).

The longest LT for *Salmonella* spp. was observed in raw cabbage (Figure 1A) among the samples, whereas the most rapid growth was observed in mung bean sprouts (Figure 1B). Across all temperatures, the highest MPD values exceeding 8 log CFU/g were consistently recorded for *Salmonella* spp. in mung bean sprouts (Figure 1C). However, *L. monocytogenes* exhibited faster growth than *Salmonella* spp. in mung bean sprouts at refrigerated temperatures due to its psychrophilic nature. The MPD value of *L. monocytogenes* was also significantly higher at 4 °C compared to other temperatures (Figure 2C), which can be attributed to reduced growth of background microbiota in mung bean sprouts at 4 °C compared to 10 °C. The predominant spoilage microorganisms in vegetables include Gram-negative bacteria such as *Pseudomonas*, *Enterobacter*, and *Flavobacterium* [41].

The study also validated the model’s suitability for predicting the behavior of different strains (isolated from various foods) and temperatures (20 and 30 °C), which were not included during the model development phase (Table 5). Overall, B*_f_*, A*_f_*, and RMSE values were found to fall within an appropriate range of 0.7 to 1.15 for the SGR and MPD of *Salmonella*. However, the model was not suitable for predicting the LT of *Salmonella* in raw and processed vegetables, likely due to physiological differences of strains isolated from various foods. On the other hand, growth models developed for cabbage salad using mixed strains of *S*. Bareilly (isolated from carrots and soybean sprouts) and *S. enteritidis* (isolated from iceberg lettuce) were confirmed to be applicable even to *S*. Typhimurium isolated from cabbage in the market. Similarly, models predicting survival and growth of *Salmonella* developed for chicken using *S.* Typhimurium showed some applicability to *S. Kentucky*, indicating similar trends, as observed in this study [42].

Regarding *L. monocytogenes* prediction models, those developed using strains isolated from enoki mushrooms (1/2b) and smoked salmon (1/2a) were found inappropriate for predicting LT when verified using strains isolated from lettuce (1/2b) and vegetables (1/2a). This inconsistency corresponds to physiological differences among isolated strains, even within the same serotype. It was confirmed that even when identical bacterial strains and serotypes are present in foods, the proliferation patterns of *L. monocytogenes* can vary depending on the food matrix. In a recent study, Ndraha et al. [15] highlighted the significant influence of strain variability in *L. monocytogenes* on its behavior, particularly within lettuce. However, the study demonstrated that MPD across all serotypes and samples showed suitable validation results, indicating no significant differences in MPD among various serotypes used for model development and validation in both raw and processed vegetables. Huang et al. [43] examined the growth trends of S. enterica and *L. monocytogenes* across five types of produce matrices under various storage temperatures, alongside their pathogenic potential in juice extracts. The study underscored that pathogen proliferation varies significantly based on pathogen species, produce types, and storage temperatures. Importantly, the research highlights the critical need for risk-based, commodity-specific food safety policies to ensure effective temperature management.

## 4. Conclusions

This study developed growth models of *Salmonella* spp. and *L. monocytogenes* in both raw (mung bean sprouts, onions, and cabbage) and processed vegetables (RTE shredded cabbage salad, cabbage juice, onion juice). It compared the growth kinetics of *Salmonella* spp. and *L. monocytogenes* and validated the models using strains isolated from different foods and temperatures, not included during the model development phase.

Overall, *Salmonella* spp. exhibited faster growth rate than *L. monocytogenes* across the various raw and processed vegetables tested. Specifically, in mung bean sprouts, *Salmonella* spp. showed faster growth at temperatures of 17 °C and above, whereas *L. monocytogenes* proliferated more rapidly under refrigeration conditions. At 8 °C, the LT and SGR values for *Salmonella* spp. in mung bean sprouts, were approximately ten times longer and three times slower, respectively, compared to those at 10 °C. In cabbage, an increase in storage temperature from 9 °C to 10 °C resulted in a more than fourfold decrease in the LT of *Salmonella* spp. The study highlights that even a slight decrease in refrigerator storage temperature by 1 or 2 degrees can significantly inhibit the growth of *Salmonella*, thereby enhancing food safety. Especially, the minimum growth temperature of *Salmonella* can vary depending on the types of vegetables and their processing methods. *Salmonella* spp. was capable of growth at 9 °C in raw onions and cabbage, at 10 °C in RTE shredded cabbage salad, and at 17 °C in cabbage juice. In contrast, *L. monocytogenes* grew at 4 °C in mung bean sprouts, raw onions, cabbage, and RTE shredded cabbage salad. Growth of *L. monocytogenes* in cabbage juice started from 17 °C, while neither pathogen showed growth in onion juice, even at 36 °C.

It should also be emphasized that the findings are specific to the mung bean sprouts, onions, cabbage, RTE shredded cabbage salad, cabbage juice, and onion juice tested in this study. Caution should be exercised when extrapolating these results to other types of vegetables or conditions. Additionally, the developed growth models can be validated with various strains, serotypes of pathogens, and other food matrices in future studies.

## Figures and Tables

**Figure 1 foods-13-02972-f001:**
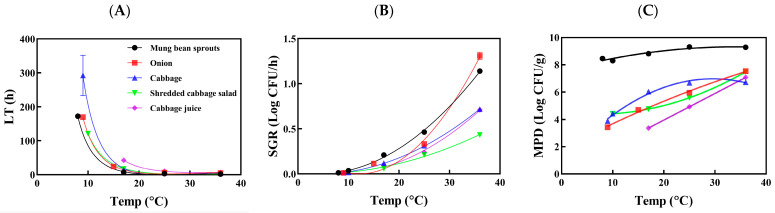
Secondary models for lag time (LT—(**A**)), specific growth rate (SGR—(**B**)), and maximum population density (MPD—(**C**)) of *Salmonella* spp. in raw and processed vegetables.

**Figure 2 foods-13-02972-f002:**
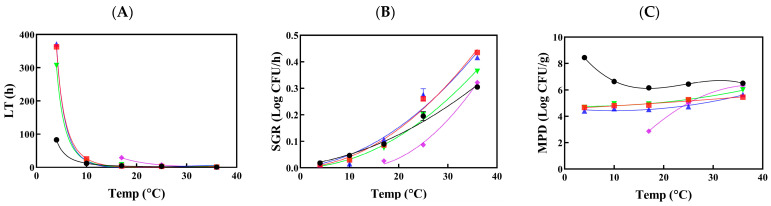
Secondary models for lag time (LT—(**A**)), specific growth rate (SGR—(**B**)), and maximum population density (MPD—(**C**)) of *L. monocytogenes* in raw and processed vegetables.

**Table 1 foods-13-02972-t001:** Analysis of native bacteria, pH, aw of raw and processed vegetables.

Sample	Microbial Populations (Log CFU/g)			pH	aw
	TAB	TCC	*E. coli*		
Mung bean sprouts	7.23 ± 0.10	6.03 ± 0.06	ND	6.30 ± 0.08	0.951 ± 0.003
Onion	2.73 ± 0.10	ND		6.30 ± 0.04	0.955 ± 0.001
Cabbage	4.67 ± 0.01			6.71 ± 0.02	0.948 ± 0.000
RTE shredded cabbage salad	2.02 ± 0.17			6.80 ± 0.02	0.962 ± 0.001
Cabbage juice	ND			5.20 ± 0.03	0.964 ± 0.001
Onion juice				4.74 ± 0.01	0.957 ± 0.001

Data are means ± standard deviation, TAB: total aerobic bacteria, TCC: total coliform counts, aw: water activity, ND: not detected, (<0.6 log CFU/g or log CFU/mL for juice).

**Table 2 foods-13-02972-t002:** Growth kinetic parameters of *Salmonella* spp. in raw and processed vegetables at various temperatures.

Sample	Storage Temperature (°C)	LT	SGR	MPD
Mung bean sprouts	7	ND	ND	ND
	8	170.1 ± 3.07 ^a^	0.012 ± 0.000 ^e^	8.47 ± 0.01 ^c^
	10	16.47 ± 1.93 ^b^	0.035 ± 0.001 ^d^	8.31 ± 0.04 ^d^
	17	7.11 ± 0.12 ^c^	0.209 ± 0.001 ^c^	8.82 ± 0.00 ^b^
	25	3.36 ± 0.28 ^d^	0.467 ± 0.025 ^b^	9.28 ± 0.13 ^a^
	36	1.85 ± 0.07 ^d^	1.138 ± 0.004 ^a^	9.30 ± 0.04 ^a^
Onion	8	ND	ND	ND
	9	169.50 ± 4.80 ^a^	0.012 ± 0.000 ^d^	3.43 ± 0.05 ^d^
	15	24.33 ± 0.26 ^b^	0.114 ± 0.004 ^c^	4.70 ± 0.07 ^c^
	25	6.18 ± 0.14 ^c^	0.332 ± 0.004 ^b^	5.96 ± 0.02 ^b^
	36	4.79 ± 0.00 ^c^	1.308 ± 0.040 ^a^	7.54 ± 0.08 ^a^
Cabbage	8	ND	ND	ND
	9	292.35 ± 59.25 ^a^	0.013 ± 0.005 ^d^	3.89 ± 0.03 ^d^
	10	68.48 ± 0.80 ^b^	0.015 ± 0.000 ^d^	4.42 ± 0.02 ^c^
	17	14.95 ± 0.18 ^c^	0.120 ± 0.000 ^c^	6.04 ± 0.06 ^b^
	25	5.34 ± 0.11 ^c^	0.312 ± 0.007 ^b^	6.69 ± 0.03 ^a^
	36	4.10 ± 0.00 ^c^	0.716 ± 0.000 ^a^	6.73 ± 0.00 ^a^
RTE shredded cabbage salad	9	ND	ND	ND
	10	121.10 ± 1.80 ^a^	0.024 ± 0.000 ^d^	4.42 ± 0.02 ^d^
	17	17.11 ± 0.69 ^b^	0.058 ± 0.000 ^c^	4.76 ± 0.03 ^c^
	25	4.46 ± 0.37 ^c^	0.210 ± 0.005 ^b^	5.59 ± 0.01 ^b^
	36	1.02 ± 0.16 ^d^	0.432 ± 0.000 ^a^	7.49 ± 0.01 ^a^
Cabbage juice	16	ND	ND	ND
	17	42.37 ± 1.11 ^a^	0.096 ± 0.005 ^c^	3.37 ± 0.01 ^c^
	25	9.65 ± 0.02 ^b^	0.240 ± 0.006 ^b^	4.93 ± 0.00 ^b^
	36	6.36 ± 0.03 ^c^	0.709 ± 0.008 ^a^	7.08 ± 0.01 ^a^
Onion juice	36	ND	ND	ND

Data are means (*n* = 4) ± standard deviation. LT: Lag time (h), SGR: Specific growth rate (log CFU/h), MPD: Maximum population density (log CFU/g or log CFU/mL for juice). ND: Not determined (no growth was observed). Different letters in the same column in each row indicate a significant difference by Duncan’s multiple range test at *p* < 0.05.

**Table 3 foods-13-02972-t003:** Growth kinetic parameters of *L. monocytogenes* in raw and processed vegetables at various temperatures.

Sample	Storage Temperature (°C)	LT	SGR	MPD
Mung bean sprouts	4	83.86 ± 1.81 ^a^	0.018 ± 0.000 ^e^	8.45 ± 0.01 ^a^
	10	13.08 ± 0.92 ^b^	0.048 ± 0.001 ^d^	6.55 ± 0.09 ^b^
	17	4.38 ± 0.05 ^c^	0.090 ± 0.001 ^c^	6.17 ± 0.01 ^d^
	25	3.13 ± 0.19 ^c^	0.187 ± 0.017 ^b^	6.38 ± 0.06 ^c^
	36	1.05 ± 0.06 ^d^	0.306 ± 0.001 ^a^	6.51 ± 0.00 ^c^
Onion	4	363.00 ± 0.50 ^a^	0.011 ± 0.000 ^e^	4.67 ± 0.06 ^d^
	10	25.37 ± 1.37 ^b^	0.030 ± 0.001 ^d^	4.81 ± 0.07 ^c^
	17	3.89 ± 0.14 ^c^	0.086 ± 0.001 ^c^	4.85 ± 0.00 ^c^
	25	2.73 ± 0.02 ^c^	0.260 ± 0.001 ^b^	5.27 ± 0.00 ^b^
	36	1.15 ± 0.12 ^d^	0.435 ± 0.007 ^a^	5.45 ± 0.07 ^a^
Cabbage	4	371.57 ± 6.31 ^a^	0.010 ± 0.001 ^d^	4.67 ± 0.06 ^d^
	10	15.37 ± 6.00 ^b^	0.030 ± 0.001 ^d^	4.81 ± 0.07 ^c^
	17	7.04 ± 0.38 ^c^	0.086 ± 0.001 ^c^	4.85 ± 0.00 ^c^
	25	4.50 ± 0.51 ^c^	0.260 ± 0.001 ^b^	5.27 ± 0.00 ^b^
	36	1.19 ± 0.00 ^c^	0.435 ± 0.007 ^a^	5.45 ± 0.07 ^a^
RTE shredded cabbage salad	4	306.77 ± 6.32 ^a^	0.012 ± 0.001 ^e^	4.68 ± 0.00 ^d^
	10	16.90 ± 1.87 ^b^	0.025 ± 0.001 ^d^	4.95 ± 0.01 ^c^
	17	7.75 ± 0.13 ^c^	0.075 ± 0.001 ^c^	4.94 ± 0.00 ^c^
	25	1.52 ± 0.38 ^d^	0.200 ± 0.004 ^b^	5.21 ± 0.03 ^b^
	36	1.04 ± 0.38 ^d^	0.364 ± 0.003 ^a^	6.01 ± 0.07 ^a^
Cabbage juice	17	29.14 ± 2.46 ^a^	0.026 ± 0.000 ^c^	2.87 ± 0.04 ^c^
	25	7.54 ± 0.92 ^b^	0.087 ± 0.003 ^b^	5.00 ± 0.18 ^b^
	36	1.06 ± 0.03 ^c^	0.322 ± 0.001 ^a^	6.34 ± 0.03 ^a^
Onion juice	36	-		

Data are means (*n* = 4) ± standard deviation. LT: Lag time (h), SGR: Specific growth rate (log CFU/h), MPD: Maximum population density (log CFU/g or log CFU/mL for juice). ND: Not determined (no growth was observed). Different letters in the same column in each row indicate a significant difference by Duncan’s multiple range test at *p* < 0.05.

**Table 4 foods-13-02972-t004:** Secondary LT, SGR, MPD models of *Salmonella* spp. and *L. monocytogenes* in raw and processed vegetables.

Pathogens	Sample	Parameter	Equation
*Salmonella* spp.	Mung bean sprouts	LT	Y = 37.37 + (−1874/T) + (23,607/T^2^)
		SGR	Y = {0.03344 × (T − 4.169)}^2^
		MPD	Y = 7.607 + (0.1005 × T) + (−0.001471 × T^2^)
	Onion	LT	Y = 53.81 + (−2592/T) + (32,676/T^2^)
		SGR	Y = {0.04756 × (T − 12.07)}^2^
		MPD	Y = 1.812 + (0.1986 × T) + (−0.001123 × T^2^)
	Cabbage	LT	Y = 93.42 + (−4713/T) + (58,501/T^2^)
		SGR	Y = {0.02666 × (T − 4.202)}^2^
		MPD	Y = 0.9006 + (0.4112 × T) + (−0.006954 × T^2^)
	RTE shredded cabbage salad	LT	Y = 32.11 + (−1836/T) + (27,241/T^2^)
		SGR	Y = {0.02014 × (T − 3.198)}^2^
		MPD	Y = 4.559 + (−0.05021 × T) + (0.003657 × T^2^)
	Cabbage juice	LT	Y = 48.97 + (−2806/T) + (45,798/T^2^)
		SGR	Y = {0.02986 × (T − 7.91)}^2^
		MPD	Y = 0.09955 + (0.1916 × T) + (0.00006041 × T^2^)
*L. monocytogenes*	Mung bean sprouts	LT	Y = 1.476 + (−35.64/T) + (1444/T^2^)
		SGR	Y = {0.01286 × (T + 7.497)}^2^
		MPD ^1^	Y = 10.54 + (−0.6337 × T) + (0.02869 × T^2^) + (−0.0003944 × T^3^)
	Onion	LT	Y = 18.33 + (−780.2/T) + (8634/T^2^)
		SGR	Y = [0.01789 × {T − (−1.381)}]^2^
		MPD	Y = 4.581 + (0.01973 × T) + (0.000143 × T^2^)
	Cabbage	LT	Y = 28.7 + (−1060/T) + (9722/T^2^)
		SGR	Y = [0.01688 × {T − (−3.122)}]^2^
		MPD	Y = 4.604 + (−0.03468 × T) + (0.001746 × T^2^)
	RTE shredded cabbage salad	LT	Y = 19.06 + (−744.7/T) + (7578/T^2^)
		SGR	Y = [0.01649 × {T − (−0.8631)}]^2^
		MPD	Y = 4.779 + (−0.009426 × T) + (0.001188 × T^2^)
	Cabbage juice	LT	Y = 8.426 + (−817.5/T) + (19,884/T^2^)
		SGR	Y = {0.02345 × (T − 11.87)}^2^
		MPD	Y = (−4.923) + (0.5887 × T) + (−0.007666 × T^2^)

T: Temperature (°C), LT: Lag time (h), SGR: Specific growth rate (log CFU/h), MPD: Maximum population density (log CFU/g or log CFU/mL for juice). ^1^ Third-order polynomial model.

**Table 5 foods-13-02972-t005:** Model performance of predictive models of *Salmonella* spp. and *L. monocytogenes* in raw and processed vegetables.

Pathogens	Sample	Parameter	B*_f_*	A*_f_*	RMSE
*Salmonella* spp.	Mung bean sprouts	LT	0.615	2.307	2.049
		SGR	0.944	1.107	0.014
		MPD	0.978	1.032	0.471
	Onion	LT	0.936	1.009	10.441
		SGR	0.754	1.388	0.023
		MPD	0.985	1.040	0.233
	Cabbage	LT	0.503	2.594	16.637
		SGR	0.934	1.156	0.014
		MPD	0.949	1.074	0.708
	RTE shredded cabbage salad	LT	1.019	1.499	1.307
		SGR	0.976	1.169	0.022
		MPD	1.006	1.009	0.085
	Cabbage juice	LT	0.920	1.103	1.873
		SGR	1.084	1.200	0.034
		MPD	1.041	1.050	0.580
*L. monocytogenes*	Mung bean sprouts	LT	1.086	1.222	0.800
		SGR	0.970	1.060	0.008
		MPD	1.010	1.021	0.222
	Onion	LT	0.757	1.952	2.098
		SGR	0.925	1.141	0.019
		MPD	0.964	1.041	0.313
	Cabbage	LT	0.858	1.765	5.675
		SGR	0.855	1.686	0.085
		MPD	1.003	1.034	0.187
	RTE shredded cabbage salad	LT	1.210	2.176	4.994
		SGR	0.944	1.249	0.026
		MPD	1.028	1.040	0.470
	Cabbage juice	LT	0.900	1.337	4.799
		SGR	1.324	1.442	0.070
		MPD	0.928	1.078	0.465

Data are means ± standard deviation. LT: Lag time (h), SGR: Specific growth rate (log CFU/h), MPD: Maximum population density (log CFU/g or log CFU/mL for juice). B*_f_*: Bias factor, A*_f_*: Accuracy factor, RMSE: Root mean square error.

## Data Availability

The original contributions presented in the study are included in the article, further inquiries can be directed to the corresponding author.

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
