# Peer review of "Modeling Behavior of Salmonella spp. and Listeria monocytogenes in Raw and Processed Vegetables"

_foods, 2024, doi:10.3390/foods13182972_

Round 1

Reviewer 1 Report

Comments and Suggestions for Authors

The author studied the growth kinetics model of Salmonella and Listeria in various raw and processing vegetables, which has certain guiding significance for the corresponding risk control. However, the overall study is not innovative enough, and there are still some problems should to be clarified in the study.

1.    Authors think these vegetables are high-risk, but the basis for confirming the statement of high microbial risk is not fully explained in the introduction.

2.    It is better to identify the serotypes of Salmonella and Listeria monocytogenes used in the study. In addition, it is necessary to explain why S. Typhimurium, S. Enteritidis and L. monocytogenes (1/2a) are selected as other Salmonella and Listeria monocytogenes strains in the cocktail inoculation for the study. In addition, the article page did not mention the significance of this kind of chicken tail mixed bacteria inoculation, because there was no separate counting, and it was impossible to judge the competitive growth of several bacteria. In addition, the article did not describe the reason for using this kind of cocktail mixed bacterial inoculation for study, since the strains were not separately counted, and it was impossible to judge the competitive growth of several bacteria.

3.    Since the contamination of TAB and TCC had been detected in some vegetables in Result 1, why not study the growth kinetics of specific pathogenic bacteria with these background bacteria, which will be more practical and instructive.

4.    The authors only chose two kinds of vegetables to be sterilized before inoculation, when other bacteria were detected except vegetable juice, which should be explained the reason?

5.    The setting of the minimum temperature of vegetables inoculated with Salmonella is different, only one degree difference. Is there any basis?

Reviewer 2 Report

Comments and Suggestions for Authors

The paper “Modeling Behavior of Salmonella spp. and Listeria monocytogenes in High-Risk Raw and Processed Vegetables” makes a valuable contribution to understanding the behavior of Salmonella spp. and L. monocytogenes in raw and processed vegetables, particularly in the context of food safety. However, the paper would benefit from clearer objectives, a more comprehensive literature review, deeper data analysis, and a more robust discussion of the study’s implications. Addressing these gaps would significantly enhance the paper’s impact and usefulness to both the scientific community and food industry practitioners.

These are my comments:

  • The objectives of the research are mentioned but could be more explicitly stated in relation to the significance of the study. The rationale behind selecting specific raw and processed vegetables for modeling microbial behavior could be better articulated. Clearly define the research objectives at the beginning of the paper and connect them directly to the broader significance of food safety in relation to the selected vegetables.
  • The methodology section is detailed but lacks specific justifications for the choice of certain models and parameters. Additionally, the criteria for the selection of strains and vegetables for the study are not fully explained. Provide justifications for the choice of models (e.g., Modified Gompertz model, Davey model) and explain why these were considered the most appropriate for the study. Also, include a detailed rationale for selecting the specific strains of Salmonella spp. and L. monocytogenes and the vegetables used.
  • The selection of temperature ranges (4°C to 36°C) for studying microbial growth is not fully justified. While the range covers common storage temperatures, the rationale for choosing these specific points, especially the extremes, is not discussed. Provide a rationale for the selection of these temperature ranges. For instance, explain why temperatures below 4°C and above 36°C were not considered, and how the chosen temperatures relate to real-world storage and distribution conditions for vegetables.
  • The methodology mentions the types of vegetables studied but does not explain the criteria used for selecting these particular vegetables (mung bean sprouts, onions, cabbage, etc.).
  • Include a justification for why these specific vegetables were chosen. Discuss their relevance in terms of their market popularity, susceptibility to contamination, or previous association with foodborne illness outbreaks.
  • The paper mentions the use of specific strains of Salmonella spp. and L. monocytogenes but does not provide detailed information on how these strains were selected, including whether they were representative of strains typically found in foodborne outbreaks. Add a detailed explanation of the criteria used for selecting these strains. Consider including information on the pathogenicity, prevalence, or resistance characteristics of these strains that make them relevant to the study.
  • The study does not describe the use of control conditions or samples. For example, there is no mention of whether non-inoculated control samples were used to account for natural microbial presence or environmental contamination during the experiments. Clarify whether control samples were used and, if not, consider including them in future studies. Controls would help to distinguish between the growth of inoculated pathogens and any natural microbial populations present in the vegetables.
  • The paper states that all experiments were conducted "at least three times," but it does not provide specific details on the number of replicates used for each experimental condition or how these replicates were statistically validated. Specify the exact number of replicates used for each condition and describe the statistical methods employed to validate the consistency and reliability of the results. This information is crucial for assessing the robustness of the study's findings.
  • The inoculation process of the vegetables with Salmonella spp. and L. monocytogenes is described, but the method used to ensure even distribution of the inoculum across the vegetable surfaces is not detailed. Provide more details on how the inoculation was carried out to ensure uniform distribution of the bacteria. This could include information on the techniques used for mixing or spreading the inoculum on the vegetable surfaces.
  • It lacks details on how these samples were handled immediately after inoculation and before storage, which could affect the initial microbial counts. Include a detailed description of the handling procedures post-inoculation and pre-storage. This should cover any steps taken to minimize contamination or temperature fluctuations that could impact the initial microbial load.
  • The paper does not mention how environmental conditions (e.g., humidity, light exposure) were controlled during the experiments, which could influence microbial growth. Discuss how environmental factors were controlled or accounted for during the experiments. If these were not controlled, suggest how they could be in future studies to ensure more reliable results.
  • The methodology describes the validation of predictive models but does not provide sufficient details on how the models were validated across different strains and conditions. Provide a more detailed explanation of the model validation process, including the specific criteria used to evaluate the models' performance and any limitations identified during validation.
  • The methodology describes pre-treatment processes such as washing and disinfection, but it does not explain how the effectiveness of these treatments was verified before proceeding with the experiments. Include a description of how the effectiveness of pre-treatment processes was assessed. For example, describe any tests conducted to ensure that disinfection processes effectively reduced the background microbial load before inoculation.
  • The interpretation of data is presented, but the discussion could benefit from a deeper analysis of the factors influencing the observed microbial behavior, particularly in different vegetable matrices and temperature conditions. Enhance the discussion by providing a more in-depth analysis of the environmental and biological factors that might have contributed to the observed differences in microbial growth across different vegetables and temperatures. Consider discussing possible limitations or variations in the data that could affect the generalizability of the results.
  • Although the paper discusses model validation, it does not thoroughly explore the limitations of the models or alternative statistical methods that could have been used. The validation process could be more robust. Discuss the limitations of the models used and consider including alternative statistical approaches that could complement or validate the findings. Additionally, expand on the model validation process, including more detailed discussions on the appropriateness and robustness of the validation metrics.
Comments on the Quality of English Language

Moderate editing.

Reviewer 3 Report

Comments and Suggestions for Authors

Summary

The authors report a modelling study of two foodborne pathogens in various vegetable products.

General Comments

Overall, the paper reports results that suggest an underlying reasonable modelling study, informed by likely adequate experimental methods.  Most critically though, some important information in missing from the methods that would be necessary to reproduce the study, such as: which specific bacterial strains are tested, how many replicates of each experiment, and what specific criteria was used to define no growth.

Other individual comments follow

Line Item Comments

L21 and 23. Here ‘superior growth’ and ‘substantially inhibits’ growth are vague.  And for L22, it’s also not clear from the data supporting faster Salmonella growth that Salmonella would be a higher risk of ‘contamination’ as contamination implies prevalence not level.  These are important scientific language issues.

L96.  Here there specific Salmonella and Listeria strains isolated from foods should be indicated.  Since there were obtained from a ministry, this implies a culture collection, so they should have some sort of ID.

Section 2.3.  Somewhere indicate the number of replicates of each growth experiment. And how replicate data were analyzed.

Table 1. ‘ND’ as not detected is less than best practice. Could you write as <#.##, where the number listed is the limited of detection of the assay. So, for example < 2.00 if the LOD is 2 log CFU/g.  Similarly, to above comment, also good to know how many reps go into the standard deviation.

L280 and Table 2. Could you expand the table to indicated the temperatures tested where growth was not observed?  Or at least somehow indicate those data.  This is because when I read around L280 my first impression of the claim that growth was not observed at a given temperature was that these were unsupported claims. This because when I looked at Table 2 I did not see an entry in the table corresponding to the temperature.  And then also indicate what specific criteria was used to claim no growth.

Figures 1 and 2.  Can you explain why some points have error bars and others do not?

470. Here you say susceptibility to Salmonella ‘contamination’ but I think you mean ‘growth’

Comments on the Quality of English Language

Overall the writing is fine.  In my above comments I did point out a few scientific English concept that could be improved.  Specifically growth v. contamination.  

Reviewer 4 Report

Comments and Suggestions for Authors

This manuscript explores the development of predictive growth models for Salmonella spp. and Listeria monocytogenes in both raw and processed vegetables under varying storage temperatures. The study focuses on raw vegetables like mung bean sprouts, onions, and cabbage, as well as processed products such as shredded cabbage salad and cabbage and onion juices. The predictive models were constructed and validated with strains isolated from different vegetable sources. I commend the authors for this comprehensive and exploratory research, which not only aims to address gaps in research by comparing contamination risks of Salmonella and Listeria across various vegetable types but also contributes to improved food safety strategies. However, to enhance the robustness of the research, I recommend the following suggestions:

Abstract

  • The abstract provides a well-structured overview of the study on microbial growth in vegetables. However, towards the end, when the authors suggest that Salmonella poses a higher risk compared to L. monocytogenes, it would be useful to include more context on why this is the case, considering that L. monocytogenes is also a serious pathogen, particularly in ready-to-eat foods.

Introduction

  • The statement "Mung bean sprouts, a staple in Korean cuisine, have seen a significant surge in domestic sales, rising from 3 million in 2020 to 141 million in 2021  does not specify the unit of measurement (e.g., kilograms, tons) and seems unrealistic without clarification. The authors should include the appropriate unit.
  • The claim that "there has been limited research comparing the likelihood of food poisoning occurrences caused by Salmonella and Listeria" might overlook existing studies. Does this statement refer to specific regions or is it a global observation? The authors should clarify.
  • The authors suggest that high-risk vegetables were identified and growth models developed but do not elaborate on how these models were validated or their predictive accuracy. This is important for assessing the reliability of the models. The authors should engage literature more thoroughly concerning this aspect in the introduction.

Methods

  • The manuscript mentions analyzing both 25 g and 10 g samples throughout, but it is unclear why different measurements were used. The authors should justify this choice.
  • The resuspension and washing steps using 0.1% peptone water do not specify how many times the washing step was performed. The authors should include this detail to ensure reproducibility.
  • The method does not provide details on how the exact concentration of each strain in the cocktail is verified. This information is crucial for reproducibility and ensuring that experimental conditions are consistent across different trials.

Results and Discussion

  • The average pH of 6.01 and aw of 0.956 are conducive to microbial growth. However, the interpretation of these values should be based on specific microbial growth thresholds.
  • Although E. coli was not detected in any of the tested samples, the absence of E. coli does not necessarily mean the absence of all pathogenic bacteria. Other undetected pathogens might still pose a risk. The authors should discuss this.
  • The recommendation for strict hygiene practices is valid, but it would benefit from specifics about which practices are most effective based on the microbial counts and conditions observed.
  • The claim that the lower pH of cabbage juice (5.2) compared to cabbage (6.71) and RTE shredded cabbage salad (6.8) explains the reduced growth of Salmonella spp. needs more detailed context. While a lower pH can inhibit microbial growth, Salmonella spp. can grow in relatively acidic conditions. The reduction in growth may also be influenced by other factors; the authors should clarify.
  • The assertion that reduced oxygen levels during the heat treatment process may have hindered the growth of pathogens needs clarification. Typically, heat treatment is used to kill microorganisms rather than inhibit growth through oxygen reduction. It would be more relevant to discuss how heat treatment affects pathogen viability rather than oxygen levels in this context.
  • Comparison with literature: Kim et al.'s report that the minimum growth temperature for L. monocytogenes is 13°C, which contrasts with observations of growth at 4°C in this study. This discrepancy should be addressed, possibly by examining differences in experimental conditions, strains, or methodologies between studies. If not, the authors should use references consistent with the study's findings.
  • The mention of MMTSO and its antimicrobial effects needs more detailed explanation. The concentrations and mechanisms should be explicitly described, and the effectiveness of MMTSO should be supported by experimental data. The specific role of MMTSO in inhibiting L. monocytogenes growth should be accurately reflected and validated.

Conclusion

  • The assertion that neither pathogen grew in onion juice even at 36°C should be consistent with the study's data. If onion juice has antimicrobial properties (like MMTSO), this should be mentioned in the conclusion to explain the lack of bacterial growth.
  • The conclusion highlights that a slight decrease in temperature significantly impedes Salmonella growth. While small temperature changes can affect microbial growth, this impact should be quantified and linked to specific findings from the study.
  • The conclusion generalizes findings to all raw and processed vegetables based on a few tested types. The authors should specify the limitations of the study's scope and acknowledge that results might not apply to all vegetable types or conditions.
  • The conclusion does not address the effectiveness or limitations of the growth models developed. It is important to discuss how well these models predicted microbial behaviour and any potential areas for improvement.

References

  • Cross-check references to ensure all cited sources are included in the reference list.
  • Ensure that references to prior studies are accurate and directly relevant to the points being made.

Comments on the Quality of English Language

Minor editing of English language required

Round 2

Reviewer 1 Report

Comments and Suggestions for Authors

The author responded effectively to all the questions raised last time. There are no other questions at present.

Reviewer 2 Report

Comments and Suggestions for Authors

/

Comments on the Quality of English Language

Minor editing